# Associations between EEG power and coherence with cognition and early precursors of speech and language development across the first months of life

Holly Bradley[1], Beth A. Smith[1,2,3]*, Ran Xiao[4]*

**1** Division of Developmental-Behavioral Pediatrics, Children's Hospital Los Angeles, Los Angeles, California, United States of America, **2** Developmental Neuroscience and Neurogenetics Program, The Saban Research Institute, Los Angeles, California, United States of America, **3** Department of Pediatrics, Keck School of Medicine, University of Southern California, Los Angeles, California, United States of America, **4** Center for Data Science, School of Nursing, Emory University, Atlanta, GA, United States of America

* bethsmit@usc.edu (BAS); ran.xiao@emory.edu (RX)

## Abstract

The neural processes underpinning cognition and language development in infancy are of great interest. We investigated EEG power and coherence in infancy, as a reflection of underlying cortical function of single brain region and cross-region connectivity, and their relations to cognition and early precursors of speech and language development. EEG recordings were longitudinally collected from 21 infants with typical development between approximately 1 and 7 months. We investigated relative band power at 3-6Hz and 6-9Hz and EEG coherence of these frequency ranges at 25 electrode pairs that cover key brain regions. A correlation analysis was performed to assess the relationship between EEG measurements across frequency bands and brain regions and raw Bayley cognitive and language developmental scores. In the first months of life, relative band power is not correlated with cognitive and language scales. However, 3-6Hz coherence is negatively correlated with receptive language scores between frontoparietal regions, and 6-9Hz coherence is negatively correlated with expressive language scores between frontoparietal regions. The results from this preliminary study contribute to the existing literature on the relationship between electrophysiological development, cognition, and early speech precursors in this age group. Future work should create norm references of early development in these domains that can be compared with infants at risk for neurodevelopmental disabilities.

## Introduction

Investigating neural development in the first months of life is critical for understanding the foundations upon which future higher-level cognitive and linguistic abilities are built. By knowing more about this period, we can gain valuable insights into infant neural maturation, informing our understanding of typical and atypical childhood development. During infancy,

**Funding:** This work was supported by the Bill & Melinda Gates Foundation [OPP1119189] (PI: BAS), website: www.gatesfoundation.org. Additionally, Dr Smith's salary was supported by National Institutes of Health grant K12-HD055929 (PI:KO), website: www.nih.gov. The funders had no role in study design, data collection and analysis, decision to publish, or preparation of the manuscript.

**Competing interests:** The authors have declared that no competing interests exist.

the human brain undergoes substantial changes [1]. Although efforts have been made to investigate and learn more about these changes, there is still much to be understood about brain development, specifically regarding human cognition and language in the first months of life. In this period, cognition encompasses a wide range of skills, including language [2]. Although cognition and language are interconnected and involve some of the same processes (namely attention, working memory, and processing speed), language is a unique skill in that it requires early attunement to native speech acoustics, such as prosody [3] and phonemes [4]. While there is evidence for some limited vocabulary knowledge at approximately 6 months of age [5], the development of language across the first months of life is governed by perceptual discrimination and sensitivity to phonotactic cues. Further, the necessity of entrenchment in social situations to acquire language sets it apart from other cognitive processes. The neural processes underpinning very early linguistic and cognitive abilities are of increasing interest and yet are still relatively understudied in the first months of life.

In this study, we investigated the correlations between resting-state (RS) electroencephalography (EEG) power and coherence with cognitive and language development longitudinally in infants approximately 1–7 months of age. Cognitive and language developmental status were assessed using the Bayley Scales of Infant and Toddler Development (3rd edition) raw cognitive, receptive language, and expressive language subscale scores. The developmental trajectory of early language acquisition is reflected in the items of the language scales in the Bayley that are tested with young infants, which mainly reflect orienting responses to speech and some very prelimited vocalizations. Thus, these items rather reflect early precursors of speech and language development, than receptive and productive language ability, as this is still very limited in the first months of life. However, in this paper, language abilities will be referred to as 'receptive' and 'expressive' as per their names on the Bayley. In this analysis, we explored the frequency band ranges of 3-6Hz and 6-9Hz. These frequency ranges were selected based on their significance as spectral landmarks in the context of our research objectives. The alpha band, traditionally defined as 8-13Hz in adults represents a prominent rhythm associated with various cognitive functions, but the precise frequency range can vary across age groups with lower frequencies prevalent in young infancy. In this study, we adopted the alpha frequency range of 6-9Hz to account for these developmental differences and align our analysis with previous research conducted in infant populations. Similarly, our choice of 3-6Hz reflective of theta processing is consistent with the range associated with certain cognitive processes such as working memory [6, 7].

EEG power and coherence are two commonly used measures in EEG research; EEG power refers to the magnitude of electrical activity at specific frequency bands, interpreted as representing the intensity of neural oscillations that can provide insights into brain function. Additionally, EEG coherence refers to the degree of synchronized activity between two or more brain regions, interpreted as quantifying the functional connectivity and information exchange among them based on the phase and amplitude relationships of the EEG signal. Using both measures together allows for a more comprehensive understanding of the cortical function underlying development. It has been noted that most of the infant RS EEG literature has solely focused on infant alpha, and how this relates to learning, cognition, and developmental outcomes, but that further inquiry into RS EEG would benefit from "investigating associations between other cortical rhythms and cognitive development", such as theta [8].

It is widely accepted that alpha band power is linked to basic cognitive processing [9]. Also, RS frontal alpha at 10 months has been shown to be predictive of executive function at 4 years of age [10]. Theta band power has also been linked to emotional and cognitive functioning skills in the second half of the first year of life, including attention modulation [11]. Further, frontal theta and social attention and executive control skills have been linked in 5-month-old

infants [12], and frontal theta power in 12-month-old infants has been used to predict language and cognitive skills at 2, 3, and 7 years of age [7]. Infants also show an increase from a baseline condition in theta power during events associated with learning or working memory [13]. Perone & Gerstein [14], using a parent-report of infant behavioral tendencies at 6 to 12 months old, found that lower levels of theta in frontal areas during baseline were associated with infant attention; namely, using another person to self-regulate. These findings provide evidence that theta and alpha band powers, specifically over the frontal cortex, are associated with cognitive and language skill development in later infancy (after 5 months of age). The literature on functional connectivity of typically developing infant EEG that reflects the dynamic interactions and coordination of neural activities of different brain regions in younger infants is relatively sparse [15]. It has been found that a greater change in left frontal alpha EEG coherence between 5 and 10 months of age was positively associated with a multitude of cognitive abilities including receptive language, attentional control, and behavioral inhibition [16]. Increased alpha coherence has been observed between 5 and 10 months of age alongside improvements in working memory performance [17, 18]. Further, in infants from 7 to 12 months of age, it has been found that increased performance on an A-not-B-task was associated with an increase in anterior/posterior coherence [19]. In infancy, RS EEG is associated with cognition and working memory, in a Piagetian A-Not-B task 8-month-old infants showed an increase in alpha coherence observed in frontal, parietal and occipital areas relative to baseline values [20].

The relationship between electrophysiological development and early language acquisition has received some attention in the past. Recent work has recorded EEG in infants at 6 months old with behavioral assessments at 6, 12, 18, 24 and 36 months of age [21]. Researchers found associations between 6 month alpha and current, but not developmental, changes in expressive language. Further, associations have been reported between newborn event related potentials (ERPs) and language and verbal memory skills at 2.5, 3.5 and 5 years old [22]. Researchers identified an 'at-risk' response pattern in the right hemisphere that was related to poorer receptive language skills at 2.5 years of age, and the same pattern in the left hemisphere was associated with poorer verbal memory skills at 5 years of age. These findings provide some evidence that ERPs can be used as potential early indicators of later language and neurocognition. It has been proposed that the developmental trajectory of language, in which infants recognize the acoustic modulations of their native language at birth and then tune into language specific patterns towards the end of the first year of life, is limited by neuronal maturation [23]. In other words, the gradual emergence of high frequency neural oscillations in infant EEG constrains language development. This is supported by the fact that maturational age is a stronger predictor of language development than ex-utero speech exposure; premature infants are potentially not able to utilize their earlier exposure to language because of electrophysiological constraints. Thus, the importance of understanding the relationship between early neural mechanisms and cognitive or linguistic development is clear.

Our goal in the present study was to complement previous studies identifying early associations between EEG measures with cognition and early precursors of speech and language development. We aimed to expand on these by assessing a novel group: young infants longitudinally over the first half year of life. Our analysis consists of an unbiased, not pre-selected, set of channels so that we can be broad and discover all potential neural candidates, as research with this age group is limited. This work has the potential to contribute to the early identification of neural function related to human cognition and speech and language development in the first half year of life. This is important because it can help us to establish a knowledge base of early brain function related to cognition and language that can be used to describe the developmental trajectories of language and cognition, and to compare with infants at risk for neurodevelopmental disabilities in future studies.

## Methodology

### Participants

EEG recordings (53 sessions) were longitudinally collected from infants with typical development (TD) (n = 21; mean age = 4.23, SD = 1.51 months at the first visit). All infants were between 38 and 203 days of age, from singleton, full-term (38 weeks minimum gestation) births, had experienced no complications during birth, and had no known visual, neurologic, or orthopedic impairment. Further, no infant scored at or below the 5[th] percentile overall for their age on the Bayley Scales of Infant and Toddler Development (3[rd] edition). Both EEG data and Bayley scores were collected in monthly increments; 3–5 sessions were acquired for 19 participants, and 1 session was acquired for 2 participants. Data were collected between February 17, 2015 and June 18, 2016.

### Procedures

The procedures for this study have previously been described in full [24]. The study was approved by the Institutional Review Board of the University of Southern California, and a parent or legal guardian provided written informed consent before participation. All methods were performed in accordance with the relevant guidelines and regulations. EEG data were acquired using a 32-electrode cap and Biosemi system at a sampling rate of 512 Hz. The data collection sessions are described as follows. First, 2 trials of 20 seconds resting-state EEG were recorded (researchers were encouraged to record 1 minute of baseline EEG here if the infant was cooperative). On average, there were 103 seconds of EEG recorded per session. For the baseline condition, a glowing and spinning globe toy was presented out of reach to the infant to maintain attention and minimize movement. Second, the infants participated in the reaching condition in which they were presented with an interesting toy at midline. Third, the toy was removed for the non-reaching condition. Reaching and non-reaching trials were alternated 5 times. Finally, the baseline condition was repeated. In addition, at each visit, the Bayley Scales of Infant and Toddler Development (3[rd] edition) was administered to the infants to measure their language, motor, and cognitive development. Of these subscores, we used the raw cognitive score (RC) as the measure of cognitive developmental status, and receptive and expressive language scores (RRL and REL) as measures for language developmental status in the study. This present study is part of a larger project investigating the development of neuromotor control during the first year of life. Other data, such as wearable motion sensor data and anthropometric data, were also recorded but were not analyzed in this study.

### Preprocessing of EEG data

Infant EEG data is susceptible to external noise and artifacts, so a series of preprocessing techniques were applied to enhance the quality of the EEG signal. All preprocessing steps were performed by using the EEGLAB toolbox (ver. 13_6_5_b) [25]. The data from all electrodes was first re-referenced to the average of T7 and T8, which helped achieve the full 80 dB common mode rejection ratio (CMRR) as recommended by BioSemi. Then a 0.3 – 30Hz bandpass infinite impulse response (IIR) filter was applied to the data. The EEG baseline conditions were extracted from the EEG recordings and visually inspected; any large fluctuations were removed. Kurtosis indices were then calculated for all electrodes, if any electrode had a Kurtosis index falling beyond 5 standard deviations of all electrodes, then it was rejected, and its signal was interpolated by surrounding electrodes. A common average reference was applied by re-referencing each electrode to the average of all electrodes to filter out common-mode artifacts. Finally, an independent component analysis was conducted to separate the baseline EEG

signal into independent components (ICs) originating from the brain source and unwanted artifacts [26]. Any components caused by electrocardiography, lateral eye movements, eye blinks, and motion artifacts were visually identified and removed to enhance the signal quality for subsequent analysis. To ensure the maximum retention of information related to brain activities, each IC was evaluated based on its temporal, spectral, and spatial features. This resulted in the exclusion of 2 to 3 ICs for most sessions.

## Spectral analysis of EEG

Power spectral densities (PSD) were estimated on the preprocessed EEG data using Welch's method [27]. The "pwelch" function in MATLAB (MathWorks Inc., Natick, MA, USA) was used for the PSD estimation. A 2-second Hann window was chosen for the PSD estimation, with a 50% overlap between segments, resulting in a 0.5 Hz frequency resolution for capturing the spectral activity changes in the infant EEG data.

To account for variation across sessions and ages and allow for comparison across all spectral activities from individual sessions, PSDs were transformed into relative powers (between 0 and 30 Hz). For each frequency bin within this range and each electrode, relative power was computed by dividing PSD by the sum of PSD from all bins. This transformation adjusted the PSDs into energy ratios within a sub-30 Hz frequency range to allow cross-session comparisons to be made. Theta and alpha relative band powers (RBP) were computed for each session by adding together all relative powers of all frequency bins within 3–6 Hz and 6–9 Hz, respectively. Further, theta and alpha power from frontal, central, and parietal brain areas were calculated by averaging key representative electrodes measuring activities in these brain regions (frontal: F3, Fz, and F4; central: C3, Cz, and C4; parietal: P3, Pz, and P4). We first investigated the correlation between RBPs of two frequency bands and infant age (in days), and the correlation between the three Bayley subscale scores and infant age (in days). Based on the obtained results (see Fig 1), we determined the infant age as a confounding factor for discovering EEG biomarkers that are strongly associated with cognitive and language development. Therefore, the partial correlation analysis was performed between RBPs at two frequency bands (i.e., theta and alpha) at those three brain regions and each of the three Bayley subscales (i.e., RC, RRL, and REL), controlling for the effect of infant age. In total, there were 18 combinations entered into the analysis to identify potential markers in EEG relative band powers associated with cognitive and language skill development. Additional procedures were implemented to adjust for multiple comparisons and investigate the impact of repeated measures in the data, which were described in detail in the statistical analysis section below.

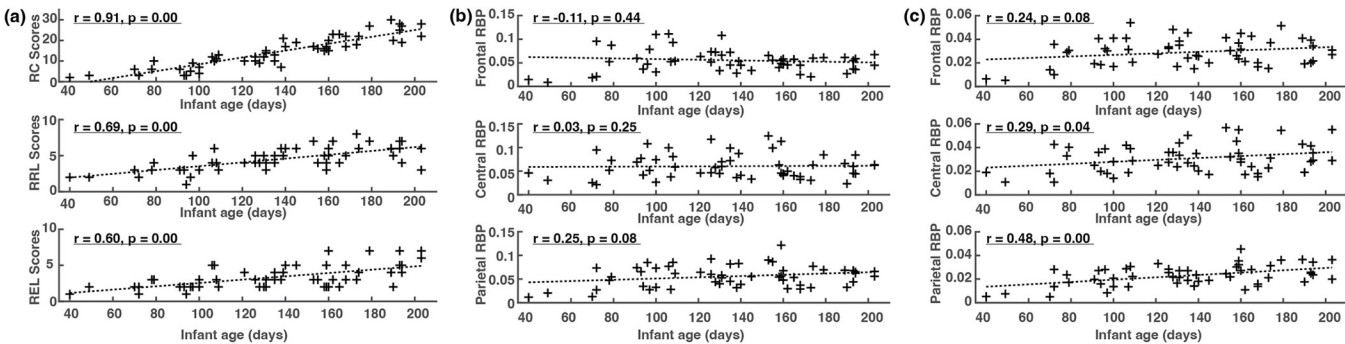

**Fig 1.** (a) Correlation between three Bayley raw subscale scores and infant age in days; (b) Correlation between theta relative band powers (RBP) and infant age in days; (c) Correlation between alpha RBPs and infant age in days. RC = Bayley raw cognitive subscale score; RRL = Bailey raw receptive language subscale score; REL = Bayley raw expressive language subscale score; RBP = relative band power.

## Connectivity analysis of EEG

To investigate the interaction and coordination of neural activities across different brain regions, we evaluated the functional connectivity by computing the magnitude-squared coherences of theta and alpha bands for 25 pairs of electrodes covering connectivity across major brain sites, including frontal, central, parietal, and occipital cortices. These electrode pairs were selected to capture both within-region and cross-region connectivity, as well as ipsilateral and bilateral coordination. The full list of 25 electrode pairs can be found in Fig 3. To calculate the magnitude-squared coherence ($Coh_{E1,E2}$) between two EEG electrodes (E1 and E2) at different frequencies ($f$), both single-electrode (i.e., auto-spectral density, $PSD_{E1}$ and $PSD_{E2}$) and cross-electrode PSD (i.e., cross-spectral density, $PSD_{E1,E2}$) were estimated by following the same parameters as the previous spectral analysis, i.e., a 2-second Hann window with 50% overlap based on Welch's method. Then, the coherence was calculated by

$$Coh_{E1,E2}(f) = \frac{(PSD_{E1,E2}(f))^2}{PSD_{E1}(f) \times PSD_{E2}(f)} \qquad \text{Eq(1)}$$

To study developmental changes in brain connectivity, we evaluated mean coherences in the theta and alpha bands from each of these 25 electrode pairs across the raw Bayley cognitive, receptive language, and expressive language subscale scores. Again, the partial Pearson's correlation was calculated for each combination of connectivity measures and Bayley subscale scores with infant age as the confounding effect. There were in total 150 combinations entered into the correlation analysis, as shown in Fig 3. Same to the spectral analysis, additional procedures were implemented to adjust for multiple comparisons and investigate the impact of repeated measures in the data, which were described in detail in the statistical analysis section below.

## Statistical analysis

To test the statistical significance of the correlations in the partial correlation analyses, the t-score was firstly calculated by

$$t = r * \sqrt[2]{\frac{n-2}{1-r^2}} \qquad \text{Eq(2)}$$

where $r$ is the sample linear partial correlation coefficient controlling for age and $n$ is the sample size in the correlation analysis ($n = 53$). Next, the p-value was calculated by comparing the t-score against the t-distribution in a two-tailed test with the null hypothesis that there is no linear correlation between the two variables in the correlation analysis. The significant level alpha was set as 0.05. When investigating the relationship between theta and alpha band powers with cognitive and language scores, average band powers from three brain regions were evaluated and the statistical tests were adjusted for multiple comparisons. Similarly, the investigation of coherences was also adjusted for multiple comparisons because of the 25 electrode pairs that entered the analysis. In all tests in the study, p values were adjusted by the Benjamini-Hochberg procedure for multiple comparisons.

Given that multiple sessions were available from different participants, there existed the issue of repeated measures for analysis in the study data that reduced the effectiveness of modeling with simple linear regression. So, a linear mixed-effects model (LMM) was adopted to account for the non-independency of data arising from participants. The LMM added random effect terms to the model to tackle non-independency, resulting in more accurate representations of outcomes [28]. The LMM models were designed as follows in Wilkinson

notation:

$$Var_{Resp} = 1 + Var_{Pred} + Age + (1|Participants) + \varepsilon \qquad \text{Eq(3)}$$

Where $Var_{Pred}$ is the predictor variable, which includes relative band powers and coherences in theta and alpha bands in the present study, with Age also included as a predictor to control for its confounding effect in the LMM analysis. $Var_{Resp}$ is the response variable, which includes the three Bayley subscale scores (i.e., RC, RRL, and REL). The models capture changes in each of the responsible variables across each of those predictor variables by considering both fixed and random effects for the slope and intercept terms. The first two elements on the right side of Eq (2) denote the fixed-effect terms, including fixed-effect intercept and slope from the EEG biomarkers and age as predictors; the term (1|$Participants$) specifies the random effects of the model that are imposed by the grouping factor, i.e., participants; $\varepsilon$ is the error term. With the LMM analysis, we investigated whether there were still strong associations between the targeted predictors (i.e., EEG biomarkers) and different cognitive and language developmental scores, even after controlling for the effect of age and accounting for repeated measures from participants in the linear mixed-effects model.

## Results

### Changes in Bayley subscale scores and EEG relative band powers along with maturation

Fig 1(A) shows significant and positive correlations between the three cognitive scores (RC, RRL, REL) and infant age in days, illustrating the development of key cognitive and language skills along with maturation. Raw cognitive scores showed a strong positive correlation with age r = 0.91, p<0.01, raw receptive language scores showed a moderate positive correlation with age r = 0.69, p<0.01, and raw expressive language scores showed a moderate positive correlation with age r = 0.60, p<0.01. Compared to those Bayley subscales, EEG relative band powers present weaker correlations with age (see Fig 1(C)), but still reached statistical significance from alpha RBPs measured from central (r = 0.29, p<0.05) and parietal lobes (r = 0.48, p<0.01).

### EEG spectral power changes along with cognitive development

Fig 2 shows the adjusted p values (after correction for multiple comparisons) for theta and alpha relative band power correlating with RC, RRL, and REL in frontal, central, and parietal brain regions. After correction for multiple comparisons, and accounting for the confounding effect of age, none of the RBPs from any frequency bands or brain regions reached statistical significance (p>0.05), failing to reject the null hypothesis that there is no significant correlation between EEG RBPs in the theta and alpha bands and the three Bayley subscale scores.

### Brain connectivity changes along with cognitive and language development

Fig 3 illustrates the adjusted p values (corrected for multiple comparisons) across all electrode pairs for theta and alpha coherences. Out of the 25 electrode pairs evaluated in this present study, one pair (F4-P4) shows significant changes in theta connectivity with respect to REL. In terms of alpha coherence, one pair (F8-P4) shows significant changes in alpha connectivity regarding RRL. All correlations are negative, illustrating decreased connectivity between frontal and parietal regions as performance increases on receptive and expressive language skills assessed by the Bayley. These findings reveal unique frontoparietal connectivity changes across language development for both theta and alpha coherences. Fig 4 shows the brain regions with

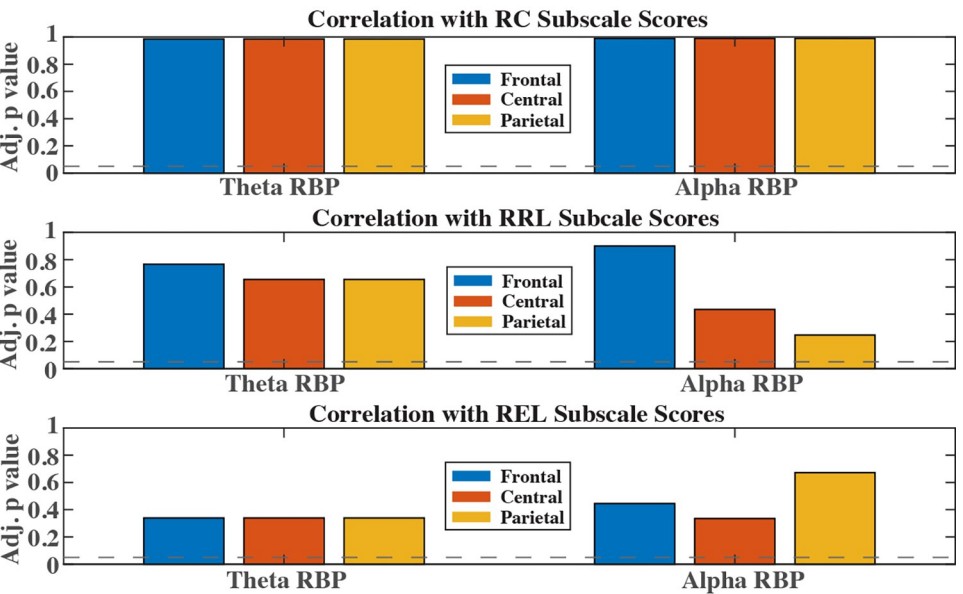

**Fig 2. Adjusted p values for Pearson's correlation between three cognitive scores and theta and alpha band relative powers.** Horizontal lines denote the significant level at 0.05.

significant connectivity changes along three Bayley scores, and topographies highlighting the relationships between brain regions with significant connectivity along two of the three Bayley subscale scores (i.e., RRL and REL). It reveals negative correlations in both theta (r = -0.45, p<0.01) and alpha (r = -0.46, p<0.01) connectivity with Bayley language subscales. And the correlations both stem from connectivity between the frontal and parietal lobes.

Table 1 presents the results of the LMM analysis, examining the associations between F4-P4 theta connectivity ($Coh_{Theta:\ F4–P4}$) and REL subscale scores, as well as between F8-P4 alpha connectivity ($Coh_{Alpha:\ F8–P4}$) and RRL subscale scores. The analysis demonstrates robust and statistically significant associations between both EEG connectivity-based markers and Bayley language subscale scores. These associations remained significant after adjusting for potential confounding factors such as infant age and accounting for repeated measures from multiple sessions involving the same participants in the study data.

## Discussion

This paper aimed to investigate early associations between EEG measures, cognition, and language. As previously noted, the Bayley divides language/communication into two categories: receptive and expressive. However, the items administered to infants in the first months of life mostly reflect early precursors of speech and language development, e.g., orienting responses to speech and limited vocalizations. While we found no significant contributions of relative band powers as predictors to the changes in cognitive and language developmental scale scores, we did find that connectivity values in frequency bands 3-6Hz and 6-9Hz were associated with early precursors of speech and language. Specifically, we found that 6-9Hz coherence (theta) is negatively correlated with receptive language scores between frontoparietal regions and that 3-6Hz coherence (alpha) is negatively correlated with expressive language scores between frontoparietal regions. The present study complements existing work in this field by demonstrating that, in the first months of life, early precursors of speech and language

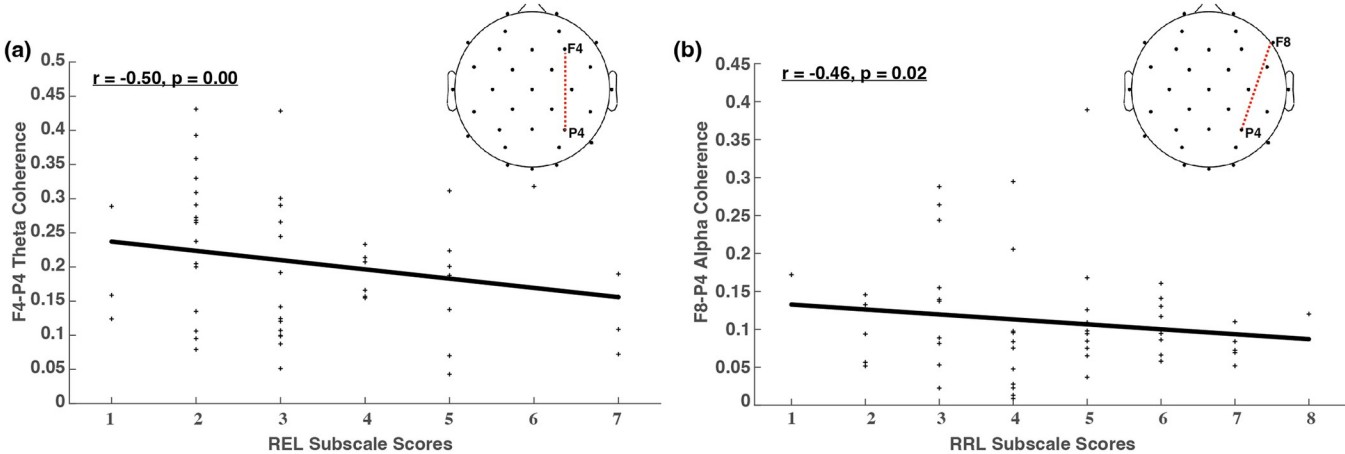

**Fig 3. Adjusted p values for correlations between cross-electrode connectivity and three Bayley subscale scores.** Each row presents the partial correlation results for one subscale (RC, RRL, and REL). Each column presents results using each frequency band (theta and alpha) for deriving the connectivity. Horizontal lines denote the significant level at 0.05. * indicates statistically significant correlation after adjustment for multiple pair comparisons and accounting for age as the confounding factor. RC = Bayley raw cognitive subscale score; RRL = Bayley raw receptive language subscale score; REL = Bayley raw expressive language subscale score.

**Fig 4. Brain regions with significant connectivity changes along Bayley language subscale scores.**

**Table 1.  Linear mixed-effect model analysis for evaluating the associations between cross-region brain connectivity and Bayley language subscales.**  * indicates statistical significance.

| LMM model | Name | t-statistics | Degree of freedom | p-value |
|---|---|---|---|---|
| $REL = 1 + Coh_{Theta:\ F4-P4} + Age + (1|Participants) + \varepsilon$ | Intercept | 1.86 | 50 | 0.07 |
| | $Coh_{Theta:\ F4-P4}$ | -4.23 | 50 | **0.00*** |
| | Age | 7.24 | 50 | 0.00* |
| $RRL = 1 + Coh_{Alpha:\ F8-P4} + Age + (1|Participants) + \varepsilon$ | Intercept | 2.49 | 50 | 0.02* |
| | $Coh_{Alpha:\ F8-P4}$ | -3.74 | 50 | **0.00*** |
| | Age | 8.41 | 50 | 0.00* |

development (both receptive and expressive) are associated with frontoparietal interhemispheric theta and alpha band connectivity values.

We did not find any significant correlations between alpha and theta EEG powers and the three Bayley subscale scores. We expected to see infant alpha band power associated with cognition as it has been shown to relate to basic cognitive processing at later ages. Most previous research reported an increase in frontal alpha associated with a cognitive task in comparison to a baseline [9, 10, 20] however, Bell (2001) found this in frontal, parietal, and occipital areas [20]. Some research has found that alpha oscillations in the parietal lobe are associated with attention [29]; for example, infants aged 8–11 months of age show attenuated alpha band power over posterior brain regions as they watch a tv show in comparison to being in total darkness. Further, in adults, it has been found that parietal alpha power is reflective of inhibition effects in the parietal attention network [30]. This present study failed to find these associations when accounting for the confounding effects of infant age, indicating that alpha band power and its association with cognitive development might emerge at later ages. This corroborates with our previous findings and other studies that indicate prominent alpha band power starts to emerge at least 6 months of age. It is also possible that high variability within and between individuals early in infancy makes it difficult to find associations. Future work should continue to explore the association between alpha band power and cognition longitudinally in the developing infant over the first year of life to investigate if relative band power may be associated with the ongoing underlying development of cognition at a certain stage in infancy. As cognitive and language abilities develop and neural functions are established, neural processing may shift towards other frequency bands or regions. Future research should also aim to investigate other frequency bands, such as beta and delta, to assess this.

In terms of coherence, we found that alpha was significantly correlated with raw expressive language scores in frontoparietal regions. Alpha coherence has been used as a measure of cognitive and language function in adults [31] and individual differences in alpha coherence in infancy have been shown to be indicative of later cognitive outcomes [17, 18]. All previous studies have investigated the development of cognition and language in infants older than 5 months of age; this present study illustrates that alpha coherence is related to infant expressive language development in the first half year of life. This channel-level EEG analysis revealed patterns suggestive of an association between increased expressive language skills and altered frontoparietal connectivity. Further, we also found that theta was significantly correlated with receptive language in frontoparietal regions, demonstrating the same relationship associating decreasing connectivity with improved receptive language performance. Together, these findings demonstrate alterations in interhemispheric frontoparietal theta connectivity with receptive language development, and interhemispheric frontoparietal alpha connectivity with expressive language development.

While it may initially seem counterintuitive that decreased frontoparietal connectivity in the right hemisphere could be associated with more advanced language development in infants at this age, we propose several potential explanations. One explanation could be hemispheric specialization; the left hemisphere is typically responsible for language processing in most individuals. Decreased right hemisphere connectivity could signify a stronger left hemisphere dominance for language processing, which is associated with more advanced language development. Conversely, certain language processes (such as prosody) may involve the right hemisphere to a greater extent. Prosodic perception, as tested by the Bayley, is a facet of complex language processing. Alterations in right hemisphere connectivity could be related to these complex language functions. Another potential explanation is that segregation allows maturation. The observed alterations may be a normal part of a typical neurodevelopmental trajectory, with the right hemisphere undergoing specific changes related to support language processing during the first months of life. The segregation of language-related functions within the right hemisphere may be a typical part of the brain's developmental trajectory, contributing to more advanced language development as it matures. All these potential explanations are speculative; more research is needed to confirm this association and understand why the decrease in frontoparietal connectivity would be associated with more advanced language development.

Limitations exist regarding this current study. This present study was a preliminary one, with only a small number of infants who were tested at varying ages over an unequal number of sessions. To clarify the existence of these associations with early language development, we will use our results to design adequately powered studies in the future. Our findings are preliminary and do not establish direct causal relationships, future studies employing more comprehensive connectivity analyses are needed to elucidate the nature and significance of these potential relationships. Despite our rigorous preprocessing efforts, preprocessing developmental EEG data inherently presents ongoing challenges. Our approach to selecting artifactual ICs was grounded in a thorough evaluation of multi-dimensional information, encompassing temporal, spatial, and spectral domains. However, there is the potential for residual artifacts to remain in the EEG data after preprocessing. This underscores the need for ongoing research and methodological advancements in the field to enhance the reliability of EEG preprocessing techniques. We also acknowledge the limitations of channel-level EEG analysis, and emphasize the need for complementary approaches, such as source localization, to investigate brain regions and connections more precisely, as channel-level EEG analysis cannot accurately infer underlying sources. Further, issues exist in that it is difficult to collect true RS EEG in infants. RS EEG studies differ quite drastically in methodology [8] with regard to condition, analysis, and brain region of interest among others which may influence the data. Infant RS EEG often involves engaging the child in a task that will keep them calm while minimizing eye and motor movements. These tasks differ from study to study but tend to involve the infant playing [32], watching a video on a screen [14], or watching a spinning globe toy, as in this study. It has been argued that the child brain does not rest in a manner that is consistent with adult brains [33], and that rest periods in infancy may actually represent periods of increased cognitive control. Future work should aim to adopt a standardized baseline EEG measure so that RS data can be more easily compared across groups and establish consistencies about what infant RS EEG really is, and how it is best measured.

To summarize, this work is the first step in identifying early associations between EEG measures with cognition and early precursors of speech and language development across the first months of life. For the next steps, we will use these preliminary results to design an adequately powered study to further investigate the relationships identified here. Future work should create norm references of early cognitive and language development. We can then use these norm

references to compare with infants at risk for neurodevelopmental disabilities. The ability to identify atypical cognitive and language development in the first months of life would support early, targeted interventions and optimal developmental outcomes.

## Author Contributions

**Conceptualization:** Holly Bradley, Ran Xiao.

**Data curation:** Holly Bradley, Ran Xiao.

**Formal analysis:** Ran Xiao.

**Funding acquisition:** Beth A. Smith.

**Investigation:** Holly Bradley, Beth A. Smith, Ran Xiao.

**Methodology:** Holly Bradley.

**Project administration:** Beth A. Smith.

**Resources:** Holly Bradley, Beth A. Smith.

**Software:** Ran Xiao.

**Supervision:** Beth A. Smith, Ran Xiao.

**Validation:** Ran Xiao.

**Visualization:** Holly Bradley.

**Writing – original draft:** Holly Bradley, Beth A. Smith, Ran Xiao.

**Writing – review & editing:** Holly Bradley, Beth A. Smith, Ran Xiao.

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
