## [Decision Letter · Decision Letter 0]

2 Feb 2024

PONE-D-23-36319Associations between EEG power and coherence with cognition and early precursors of speech and language development across the first months of lifePLOS ONE

Dear Dr. Smith,

Thank you for submitting your manuscript to PLOS ONE. After careful consideration, we feel that it has merit but does not fully meet PLOS ONE’s publication criteria as it currently stands. Therefore, we invite you to submit a revised version of the manuscript that addresses the points raised during the review process.

Two experts in the field have carefully reviewed the manuscript entitled, "Associations between EEG power and coherence with cognition and early precursors of speech and language development across the first months of life". Their comments are appended below.

Both reviewers acknowledged the manuscript is fairly well written, however, there are several critical concerns which should be avoided publication as it stands.  I am sure each comment would strengthen the manuscript. I would suggest revising according to the comments.

We look forward to receiving your revised manuscript.

Kind regards,

Manabu Sakakibara, Ph.D.

Academic Editor

PLOS ONE

 [This work was supported by the Bill & Melinda Gates Foundation [OPP1119189] (PI: BAS), website www.gatesfoundation.org. Additionally, Dr. Smith’s salary was supported by National Institutes of Health grant K12-HD055929 (PI: KO), website: www.nih.gov.].  

[This work was supported by the Bill & Melinda Gates Foundation [OPP1119189] (PI: BAS), website www.gatesfoundation.org. Additionally, Dr. Smith’s salary was supported by National Institutes of Health grant K12-HD055929 (PI: KO), website: www.nih.gov]

 [This work was supported by the Bill & Melinda Gates Foundation [OPP1119189] (PI: BAS), website www.gatesfoundation.org. Additionally, Dr. Smith’s salary was supported by National Institutes of Health grant K12-HD055929 (PI: KO), website: www.nih.gov.]

5. Please include the reference section of your manuscript.

Reviewers' comments:

Reviewer's Responses to Questions

**Comments to the Author**

1. Is the manuscript technically sound, and do the data support the conclusions?

Reviewer #1: Yes

Reviewer #2: Yes

2. Has the statistical analysis been performed appropriately and rigorously? 

Reviewer #1: Yes

Reviewer #2: Yes

3. Have the authors made all data underlying the findings in their manuscript fully available?

Reviewer #1: Yes

Reviewer #2: Yes

4. Is the manuscript presented in an intelligible fashion and written in standard English?

Reviewer #1: Yes

Reviewer #2: Yes

5. Review Comments to the Author

Reviewer #1: This paper the Authors investigated EEG power and coherence in infancy, and their relations to cognition and early recursors of speech and language development.

The paper is well-written and interesting. Methodology is clearly described, with all necessary information, such as the explanation about EEG recordings and their pre-processing steps.

Furthermore, relevant statistical measures are used by the Authors to test the statistical significance. Discuss is well-written, with possible disadvantages of the research and future directions.

I proposed accepting the paper with some minor suggestions as follows:

1. Page 4 - different font detected inside text.

2. Page 9 - after Eq. 2, should r and n be italic as in corresponding Equation 2? If yes, check the whole paper for similar errors.

3. Resolution of Figures can be improved (such as Figure 1). Especially if as large as currently in the paper.

Reviewer #2: The present study complements existing work with the aim of researching the early precursors of speech and language development across the first months of life through the analysis of EEG power and coherence in two significant frequency bands (3-6 Hz, Theta and 6-9Hz, Alpha).

I only have one concern regarding the way the work was conducted and was described in the preprocessing phase of the EEG data. There are some aspects on which I invite the authors to reflect.

In the paragraph 2.3 it is written that you use ica for removing artifacts. Several things have not been specified in this regard: the type of algorithm used, the length in time of the EEG signal to which you apply the IC, the number of components into which the EEG signal is decomposed and finally the sampling frequency of the acquired signal. This is all useful information to understand if the ICA has been used correctly since for example if the recorded signal is too short and was acquired with a sampling frequency that is not too high (below 1024 Hz) it is not It is always possible to break down the signal into a number of components equal to the number of electrodes because the information content is not sufficient (the number of points in which the signal was sampled for the length of the signal in time). The risk of using IC in an improper way is that mixed components are removed, i.e. those that contain both artefactual and cerebral signals because the software has not been able to separate the components adequately. Furthermore, information was omitted regarding the classification method of the artefactual components: manual or automatic? It should be noted that the most common brain signal artifact removal systems used on adult EEG signals are not always effective on children's EEG signals, especially on brain signals from children in their first year of age.

6. PLOS authors have the option to publish the peer review history of their article (what does this mean?). If published, this will include your full peer review and any attached files.

Reviewer #1: No

Reviewer #2: No

---

## [Author Response · Author response to Decision Letter 0]

16 Feb 2024

Response to reviewers

We express our sincere gratitude to the reviewers for their constructive comments and valuable feedback, which have significantly contributed to the enhancement of our study. We have revised the manuscript to consider all remarks from the reviewers. We have uploaded the revised manuscript with text changes marked in red color and formatting changes identified using track changes. Please find our point-by-point responses below.

Reviewer #1: 

In this paper the Authors investigated EEG power and coherence in infancy, and their relations to cognition and early precursors of speech and language development. The paper is well-written and interesting. The methodology is clearly described, with all the necessary information, such as the explanation about EEG recordings and their pre-processing steps. Furthermore, relevant statistical measures are used by the Authors to test the statistical significance. Discuss is well-written, with possible disadvantages of the research and future directions. I proposed accepting the paper with some minor suggestions as follows:

1. Page 4 - different font detected inside text.

We deeply appreciate the reviewer’s comment that helps improve the manuscript. We have fixed the inconsistent font issue, which is caused by a glitch in the reference manager. We have also gone through the manuscript to address similar issues.

2. Page 9 - after Eq. 2, should r and n be italic as in corresponding Equation 2? If yes, check the whole paper for similar errors.

We have revised the font style of these notions. We have also gone through the manuscript to make sure similar issues have been addressed. 

3. Resolution of Figures can be improved (such as Figure 1). Especially if as large as currently in the paper.

We have taken steps to enhance the resolution of all figures within the manuscript, ensuring they are of high quality and legibility.

Reviewer #2: 

The present study complements existing work with the aim of researching the early precursors of speech and language development across the first months of life through the analysis of EEG power and coherence in two significant frequency bands (3-6 Hz, Theta and 6-9Hz, Alpha). I only have one concern regarding the way the work was conducted and was described in the preprocessing phase of the EEG data. There are some aspects on which I invite the authors to reflect.

In the paragraph 2.3 it is written that you use ICA for removing artifacts. Several things have not been specified in this regard: the type of algorithm used, the length in time of the EEG signal to which you apply the IC, the number of components into which the EEG signal is decomposed and finally the sampling frequency of the acquired signal. This is all useful information to understand if the ICA has been used correctly since for example if the recorded signal is too short and was acquired with a sampling frequency that is not too high (below 1024 Hz) it is not It is always possible to break down the signal into a number of components equal to the number of electrodes because the information content is not sufficient (the number of points in which the signal was sampled for the length of the signal in time). The risk of using IC in an improper way is that mixed components are removed, i.e. those that contain both artefactual and cerebral signals because the software has not been able to separate the components adequately. Furthermore, information was omitted regarding the classification method of the artefactual components: manual or automatic? It should be noted that the most common brain signal artifact removal systems used on adult EEG signals are not always effective on children's EEG signals, especially on brain signals from children in their first year of age.

We greatly appreciate the reviewer's insightful comment regarding the preprocessing challenges of infant EEG data. Recognizing the complexity of this issue, we have approached the preprocessing with utmost care, aiming to enhance signal quality while preserving essential brain activities. We adopted a conservative stance on artifact removal, ensuring that our practices minimize the risk of losing valuable data. In our study, each session had an average of 103 seconds of 32-channel EEG recordings, sampled at 512 Hz. This provides sufficient data points for robust ICA analysis, aligning with the empirical guidelines (Onton et al., 2006). We employed a meticulous manual inspection, evaluating each IC's temporal, spectral, and spatial characteristics. This comprehensive approach allowed us to distinguish and reject those components most likely to represent artifacts. Despite our rigorous preprocessing efforts, we acknowledge the inherent challenge of eliminating residual artifacts in developmental EEG, particularly given that the majority of existing preprocessing techniques are primarily developed for adult EEG analysis, a limitation now explicitly mentioned in our discussion section.

Onton J, Westerfield M, Townsend J, Makeig S. Imaging human EEG dynamics using independent component analysis. Neurosci Biobehav Rev. 2006;30(6):808-22

---

## [Decision Letter · Decision Letter 1]

27 Feb 2024

Associations between EEG power and coherence with cognition and early precursors of speech and language development across the first months of life

PONE-D-23-36319R1

Dear Dr. Smith,

We’re pleased to inform you that your manuscript has been judged scientifically suitable for publication and will be formally accepted for publication once it meets all outstanding technical requirements.

Kind regards,

Manabu Sakakibara, Ph.D.

Academic Editor

PLOS ONE

Additional Editor Comments (optional):

Reviewers' comments:

Reviewer's Responses to Questions

**Comments to the Author**

1. If the authors have adequately addressed your comments raised in a previous round of review and you feel that this manuscript is now acceptable for publication, you may indicate that here to bypass the “Comments to the Author” section, enter your conflict of interest statement in the “Confidential to Editor” section, and submit your "Accept" recommendation.

Reviewer #1: All comments have been addressed

2. Is the manuscript technically sound, and do the data support the conclusions?

Reviewer #1: Yes

3. Has the statistical analysis been performed appropriately and rigorously? 

Reviewer #1: Yes

4. Have the authors made all data underlying the findings in their manuscript fully available?

Reviewer #1: Yes

5. Is the manuscript presented in an intelligible fashion and written in standard English?

Reviewer #1: Yes

6. Review Comments to the Author

Reviewer #1: Dear Authors, thank you for addressing my comments. I have no further suggestions. I suggest accepting this paper.

7. PLOS authors have the option to publish the peer review history of their article (what does this mean?). If published, this will include your full peer review and any attached files.

Reviewer #1: No

---

## [Editor Report · Acceptance letter]

29 Feb 2024

PONE-D-23-36319R1 

PLOS ONE

Dear Dr. Smith, 

I'm pleased to inform you that your manuscript has been deemed suitable for publication in PLOS ONE. Congratulations! Your manuscript is now being handed over to our production team.

Kind regards, 

on behalf of

Dr. Manabu Sakakibara 

Academic Editor

PLOS ONE